# Rotator Cuff Tenocytes Differentiate into Hypertrophic Chondrocyte-Like Cells to Produce Calcium Deposits in an Alkaline Phosphatase-Dependent Manner

**DOI:** 10.3390/jcm8101544

**Published:** 2019-09-26

**Authors:** Christelle Darrieutort-Laffite, Paul Arnolfo, Thomas Garraud, Annie Adrait, Yohann Couté, Guy Louarn, Valérie Trichet, Pierre Layrolle, Benoit Le Goff, Frédéric Blanchard

**Affiliations:** 1INSERM UMR1238, Bone Sarcoma and remodeling of calcified tissues, Nantes University, 44093 Nantes, France; paul.arnolfo@chu-nantes.fr (P.A.); thomas.garraud@chu-nantes.fr (T.G.); valerie.trichet@univ-nantes.fr (V.T.); pierre.layrolle@univ-nantes.fr (P.L.); benoit.legoff@chu-nantes.fr (B.L.G.); frederic.blanchard@univ-nantes.fr (F.B.); 2Rheumatology department, Nantes University Hospital, 44093 Nantes, France; 3Univ. GrenobleAlpes, CEA, INSERM, IRIG, BGE, F-38000 Grenoble, France; annie.adrait@cea.fr (A.A.); yohann.coute@cea.fr (Y.C.); 4Institut des Matériaux Jean Rouxel (IMN) - UMR CNRS 6502, Nantes University, 44300 Nantes, France; guy.louarn@univ-nantes.fr

**Keywords:** calcific tendonitis, rotator cuff, apatite, hypertrophic chondrocytes, tissue non-specific alkaline phosphatase

## Abstract

Calcific tendonitis is a frequent cause of chronic shoulder pain. Its cause is currently poorly known. The objectives of this study were to better characterize the cells and mechanisms involved in depositing apatite crystals in human tendons. Histologic sections of cadaveric calcified tendons were analyzed, and human calcific deposits from patients undergoing lavage of their calcification were obtained to perform infrared spectroscopy and mass spectrometry-based proteomic characterizations. In vitro, the mineralization ability of human rotator cuff cells from osteoarthritis donors was assessed by alizarin red or Von Kossa staining. Calcifications were amorphous areas surrounded by a fibrocartilaginous metaplasia containing hypertrophic chondrocyte-like cells that expressed tissue non-specific alkaline phosphatase (TNAP) and ectonucleotide pyrophosphatase/phosphodiesterase 1 (ENPP1), which are two key enzymes of the mineralization process. Calcific deposits were composed of apatite crystals associated with proteins involved in bone and cartilage development and endochondral bone growth. In vitro, tenocyte-like cells extracted from the rotator cuff were able to mineralize in osteogenic cultures, and expressed *TNAP*, *type X COLLAGEN*, and *MMP13*, which are hypertrophic chondrocytes markers. The use of a TNAP inhibitor significantly prevented mineral deposits. We provide evidence that tenocytes have a propensity to differentiate into hypertrophic chondrocyte-like cells to produce TNAP-dependent calcium deposits. We believe that these results may pave the way to identifying regulating factors that might represent valuable targets in calcific tendonitis.

## 1. Introduction

Calcific tendonitis is a frequent cause of chronic shoulder pain. Apatite deposits present within the rotator cuff tendons lead to subacromial impingement. Although frequent, with 10% to 42% cases of chronic shoulder pain [1], its etiology remains unknown, and only little data is available on the mechanisms leading to calcium deposits. Histological analysis showed the presence of a fibrocartilaginous metaplasia around the deposits containing chondrocyte-like cells [2]. However, the role that these cells play in mineralization has not been clearly determined. Uhthoff et al. observed alkaline phosphatase activity in chondrocyte-like cells and in the surrounding matrix, suggesting that the phenomenon could be incomplete endochondral ossification through the coalescence of calcium crystals present in matrix vesicles [2]. In contrast, Archer et al. did not detect any alkaline phosphatase enzymatic activity in areas containing the rounded cells, and refuted the previous hypothesis [3]. In calcific tendonitis, deposits are not linked to the dysregulation of calcium and phosphorus metabolism [4], as opposed to those observed in patients with chronic kidney disease. In the latter, heterotopic calcifications are more extensive and involve other tissues than the tendons (subcutaneous, muscles). 

Tenocytes are highly specialized mesenchyme-derived cells responsible for the synthesis and maintenance of dense regular connective tissue. They are embedded in a three-dimensional network of extracellular matrix components composed of type I collagen (60–85% of the dry weight), other collagen subtypes (III and V), proteoglycans, glycosaminoglycans, and glycoproteins [5]. When isolated and expanded in vitro, these cells are named tenocyte-like cells (TLC). They are characterized by the expression of a combination of markers: collagens I and III, decorin, tenomodulin, scleraxis, Mohawk homeobox (Mkx), and tenascin-C [6]. TLC also express mesenchymal stem cell surface markers, and have the ability to differentiate into adipogenic, osteogenic, and chondrogenic phenotypes [7]. Some data suggested that multipotential stem/progenitor cells are present in tendons and are implicated in tendon regeneration [8,9]. It has not yet been determined which cells may be responsible for the calcific deposits.

The objectives of this study were, in the first section, to describe the organization of the calcium deposits within the tendon and to characterize the cells observed around them, and in the second, to study the ability of cells extracted from the rotator cuff to induce mineralization in vitro, the phenotype of these cells, and the mechanisms involved. In the last section, we analyzed the composition of calcium deposits extracted from patients suffering from calcific tendonitis.

## 2. Experimental Section

### 2.1. Histology and Immunochemistry 

Rotator cuff tendons were collected at the Nantes anatomical facilities from fresh and non-embalmed cadaveric subjects kept at +4 °C. Approval was obtained from the local Institutional Review Board and Ethics Committee for the use of human anatomical specimens. Ultrasound (US) was first used to detect calcified tendons. Tendons were considered as normal when they had a continuous, fibrillary, and hyperechoic structure on US. One normal and five calcified tendons were collected from five subjects (median age 82 years (54–88 years), three males). Of the five calcifications, three had an arc-shaped morphology, and two were fragmented without acoustic shadowing. After fixation, they were scanned [10], decalcified [11], and embedded. Three μm-thick sections were stained with hematoxylin and eosin (HE), alcian blue, Safranin O/Fast Green (SOFG) and tartrate-resistant acid phosphatase (TRAP) [12,13]. Von Kossa was performed on a not decalcified portions of the samples.

Sections were incubated with primary antibodies targeting RUNX2 (Novus Biologicals NBP1-77461, 1/200), SOX9 (ab3697, 1/400, Abcam, UK), Type II collagen (COL2A1) (sc-52658, 1/1000, Santa Cruz Biotechnology, USA), Type X collagen (COL10) (ab49945, 1/1000, Abcam), tissue non-specific alkaline phosphatase (TNAP) (ab108337, 1/1000, Abcam), ectonucleotide pyrophosphatase/phosphodiesterase 1 (ENPP1) (ab40003, 1/2000, Abcam), Mohawk homeobox (MKX) (NBP2-45863, 1/400, Novus Biologicals, USA), CD31 (ab28364, 1/100, Abcam), CD68 (M0876, 1/400, Dako, USA), and cleaved caspase-3 (9664, 1/400, Cell Signaling, Netherlands). Secondary antibodies at 1:400 dilutions were incubated for 2 h and streptavidin-HRP (P0397, Dako) was incubated for 1 h, both at room temperature. Staining was achieved with DAB (TA-125-QHDX, Thermo Scientific) and counterstaining was achieved with hematoxylin. Cells were considered positive when stained in brown. 

### 2.2. Cell Culture 

*Tenocyte-like cells (TLC).* Rotator cuff tendon samples (biceps or supraspinatus) from 11 donors (10 females) were obtained surgically at the time of shoulder replacement surgery for osteoarthritis (six for primitive osteoarthritis (OA), four for rotator cuff tear arthropathy, and 1 for OA secondary to osteonecrosis of the humeral head). Prior to sample collection, the absence of tendon calcification was checked on the X-ray of the shoulder. All donors enrolled gave their formal consent. The study was approved by the local ethics committee and by the French Research Ministry (DC-2011-1399) in accordance with the Declaration of Helsinki. Mean age was 72.1 years (± 6.5). Tendons were digested in collagenase A (1 mg/mL) [6,7,14]. Cells were cultured in RPMI medium (Lonza, Switzerland) supplemented with 10% fetal bovine serum (ThermoScientific), as well as essential and non-essential amino acids (Sigma-Aldrich, Germany), and used until passage 2. TLC were characterized by flow cytometry (supplemental methods).

*Osteogenic medium.* TLC were incubated in osteogenic medium containing dexamethasone 10^−7^ M, *β*-glycerophosphate 10 mM, and ascorbic acid 2P 250 µM [15]. A tissue non-specific alkaline phosphatase inhibitor (TNAPi) was used at 20 µM (EMD Millipore Corp, USA, ref 613810-10MG). Alizarin red S staining was performed [15], and TNAP activity was quantified using an alkaline phosphatase Colorimetric Assay Kit (Abcam) after cell lysis [16].

*Chondrogenic three-dimensional culture.* To obtain pellets, cells were detached from the culture flask with Trypsin-EDTA (Lonza) and after centrifugation, they were counted and resuspended to generate a cell solution of 1.5 × 10^7^ cells/mL in StemPro Osteocyte/Chondrocyte Differentiation basal medium. Five-µL droplets of cell solution were seeded in the center of wells (24-well plate), and pellets were then incubated during 2 h at 37 °C, 5% CO_2_. After 2 h, complete StemPro Chondrogenesis Differentiation Medium (A10064-01, Gibco, USA) was added and changed every 2–3 days during a week. For the three following weeks, the pellets were maintained in the chondrogenic medium (CH condition) or received a mixed chondrogenic/osteogenic medium (chondrogenic medium with dexamethasone 10^−7^ M, *β*-glycerophosphate 10 mM, and ascorbic acid 2P 250 µM (CH-OM condition)). Chondrogenic medium was combined with the osteogenic molecules because without the chondrogenic medium, cells re-adhered in a monolayer (data not shown). One week in chondrogenic medium was necessary to obtain well-formed pellets in culture, to which we added three weeks of culture with osteogenic molecules (same duration as 2D culture).

### 2.3. RT-qPCR Analysis

Tenocyte total RNA was extracted using mini spin columns (NucleoSpin^®^ RNA, Macherey Nagel, France). The reverse transcription (RT)-qPCR was carried out as described [17]. The primers used are reported in Appendix A. Resultant cycle threshold (Ct) values were normalized to the invariant control, *GAPDH* (glyceraldehyde-3-phosphate dehydrogenase), and expressed as 2^−ΔCt^.

### 2.4. Scanning Electron Microscopy (SEM) and FTIR (Fourier Transform Infrared) Spectroscopy

Calcifications were extracted from 14 patients undergoing an ultrasound-guided percutaneous lavage in our Rheumatology department (mean age 52.4 years (±12.3), 7 men). All patients enrolled gave their formal consent. Calcific deposits were collected in saline, washed in phosphate-buffered saline (PBS), and dehydrated in ethanol 70%. SEM was performed with a TM300 microscope (Hitachi, Japan), and FTIR spectroscopy was performed using a Bruker VERTEX 70 spectrometer with an ATR accessory equipped with a diamond crystal and using a range of 400–4000 cm^−1^. 

### 2.5. Proteomic and Bioinformatics Analysis

Calcifications from 5 patients were incubated for 2 h in RIPA buffer, protease inhibitors (leucopeptine (0.4 µg/mL), aprotinine (0.4 µg/mL)), phenylmethylsulfonyl fluoride (PMSF) (5 µM), and NaVO4 (0.5 mM). Extracted proteins were solubilized in Laemmli buffer, heated for 5 min at 95 °C, before being stacked at the top of a SDS-PAGE gel (4–12% NuPAGE, Life Technologies), stained with Coomassie blue R-250 and in-gel digested using modified trypsin (Promega, sequencing grade), as previously described [18]. The resulting peptides were analyzed by online nanoliquid chromatography coupled with tandem MS (UltiMate 3000 and LTQ-Orbitrap Velos Pro, Thermo Scientific, USA). Peptides were sampled on a 300 µm × 5 mm PepMap C18 precolumn and separated on a 75 µm x 250 mm C18 column (PepMap, Thermo Scientific) using a 120-min gradient. MS and MS/MS data were acquired using Xcalibur (Thermo Scientific). Peptides and proteins were identified using Mascot (version 2.6) through concomitant searches against Uniprot (Homo sapiens taxonomy, December 2018 version), a classical contaminant database (homemade), and the corresponding reversed databases. The Proline software (http://proline.profiproteomics.fr) was used to filter the results: a conservation of rank 1 peptides, a peptide identification false discovery rate <1% as calculated on peptide scores by employing the reverse database strategy, a minimum peptide score of 25, and a minimum of one specific peptide per identified protein group. Proline was then used to perform a compilation, grouping, and spectral counting-based comparison of the protein groups identified in the different samples. Proteins from the contaminant database and keratins were discarded from the final list of identified proteins. Only protein groups identified with a total of at least two specific spectral counts in total were retained. 

The proteins identified in our proteomic analysis of calcifications were subjected to functional classifications using PANTHER v.14.0 [19]. 

### 2.6. Statistics

Results were analyzed with the Mann–Whitney test using GraphPad8 software. Data are given as mean ± SEM and results with *p* < 0.05 were considered significant. For proteomic analyses, the protein list was explored for statistical over-representation of GO (Gene Ontology) biological process and cellular component terms using Fisher’s exact test and the Bonferroni correction for multiple testing. Only GO terms with an adjusted *p*-value < 0.05 were considered as significant hits.

## 3. Results

### 3.1. Histological Characterization of Calcific Tendonitis

To characterize the cell and tissue organization around calcific deposits, we analyzed samples collected from cadaveric supraspinatus tendons. Micro-computed tomography (CT) images showed that the calcifications were composed of multiple adjacent deposits. Their density was higher when an arc-shaped morphology was observed with ultrasound (donors 3, 4, and 5) (Figure 1A). Von Kossa staining confirmed the presence of small micrometric and large millimetric calcified areas in the five donors (Figure 1C: example of a large calcification), but not in the normal tendon (Appendix A). In the decalcified samples, mineral deposit sites appeared as areas containing an amorphous acellular material that remained after decalcification (Figure 1B). Four donors had small (vesicular, subcellular-sized) calcifications disseminated between tendinous fibers in association with nearby voluminous ones encapsulated by a tissue. In one sample (donor 5), an intratendinous osseous metaplasia with osteoblasts, osteocytes, and bone marrow was observed adjacent to the large encapsulated calcium deposits (Figure 1B). The capsule presented a red coloration after SOFG staining (proteoglycan specific), and a blue coloration with alcian blue staining (mucopolysaccharides specific), which was indicative of fibrocartilaginous tissue. This fibrocartilaginous metaplasia was not observed within the non-calcified part of the samples or in the normal tendon. However, this tissue was COL2 negative on immunohistochemistry, whereas the fibrocartilage at the enthesis was strongly positive, as expected (Figure 1C). 

Within the fibrocartilaginous area surrounding the deposits, cells with rounded nuclei and pericellular lacunae were observed, suggesting chondrocyte differentiation. These cells expressed the master transcription factors RUNX2 and SOX9, the enzymes ENPP1 and TNAP, and some of them expressed type X collagen (COL10), which was indicative of hypertrophic chondrocyte differentiation (Figure 1D,E). Interestingly, extracellular TNAP staining was also present at the interface between the calcium deposits and the fibrocartilaginous metaplasia, suggesting an active role of this enzyme in apatite deposition. In contrast, tenocytes observed in the non-calcified part of the samples and in the normal tendon did not express these markers. 

We also identified blood vessels (CD31+) surrounding the deposits in four of the five calcified samples. The median of the smallest distance was 56.5 µm (Figure 1E). In the non-calcified portion of the samples, we did not observe such vascularization, with normal vascularization being located in the sub-acromial bursae. 

We did not observe CD68+ macrophages, multinucleated giant cells, or TRAP-positive cells (i.e., osteoclasts) around the deposits (data not shown). Pathological calcification in cartilage can be associated with chondrocytes apoptosis [20], but none of the chondrocyte-like cells expressed cleaved caspase-3 (data not shown). 

### 3.2. Characterization of Tendon Cells and Assessment of their Mineralization Ability

Cells isolated from rotator cuff tendons had a spindle shape and expressed *COL1A1, COL3A1, MKX,* and *SCLERAXIS,* as expected. However, *TENOMODULIN* was very weakly expressed and lost after passage 1 (Appendix A). The majority of cells expressed CD44, CD73, CD90, and CD105, and were negative for CD45, CD133, and CD146 (Appendix A). These cells were thus termed tenocyte-like cells (TLC) [7].

After 21 days of TLC culture in the osteogenic medium, we obtained significant mineral deposits compared to the control medium (Figure 2A, *N* = 5). Mineral deposits occupied 15.48% (± 2.57) of the surface of the well for the osteogenic condition versus 0.50% (± 0.4) for the control medium. As a positive control, human mesenchymal stem cells mineralized 53.33% (± 7.07) of the total surface of the well (data not shown).

### 3.3. Phenotype of the Mineralizing TLC

Total RNA was extracted from TLC after 21 days of culture, and the expression of several genes was measured (Figure 2B). In osteogenic medium, we observed significantly increased expression of *TNAP* and *ENPP1*, which are both implicated in the mineralization process. We also observed an increased expression of several chondrocyte markers, including *COL10A1,* matrix metallopeptidase 13 (*MMP13*)*,* and cartilage oligomeric matrix protein (*COMP*), which are especially expressed by hypertrophic chondrocytes, but *COL2A1* was not detected, and the expression of *ACAN* was significantly reduced. The osteogenic medium did not have any incidence on the expression of the other genes related to osteoblast or tenocyte differentiation. 

### 3.4. TNAP Involvement in TLC-Induced Mineralization 

We observed a significant increase in TNAP activity from day 14 onward in osteogenic medium (Figure 3A). The use of a TNAP inhibitor showed a significant reduction in the mineral deposits induced by the osteogenic medium (Figure 3B,C), suggesting a critical role of TNAP in the process. The TNAPi modified the phenotype of the cells and induced a decrease in the expression of *COL10* and *MMP13*, which is usually expressed by hypertrophic chondrocytes (Figure 3D). However, it did not affect the expression of the other genes studied: *TNAP, ENPP1, ACAN,* and *COMP*. We did not observe any effect on the expression of *SOX9, RUNX2, SPP1, COL1A1, COL2A1,* or *SCX* (data not shown).

### 3.5. Mineralization Ability of TLC in 3D Chondrogenic Culture

We next wondered whether a direct chondrogenic stimulation of TLC would also favor mineral deposits. First, TLC 3D pellets cultured in a chondrogenic medium for four weeks were able to produce peripheral fibrocartilaginous tissue with an abundant matrix positive for alcian blue staining, but negative for COL2 (Figure 4A). The mean diameter of the pellets was 439 µm (± 54 µm). The center of the pellets was much less structured, with a poor extracellular matrix. This could be explained by an impaired diffusion of the soluble factors present in the medium with limited effects, as we observed previously with chemotherapeutic agents [21]. The cells observed in the fibrocartilaginous tissue had an elongated fibroblastic shape. By immunohistochemistry, these cells expressed osteoblast and chondrocyte markers such as SOX9, RUNX2, COL10, TNAP, and ENPP1. They also expressed MKX, which is a tenocyte marker. RT-qPCR performed on the whole pellets confirmed the expression of *RUNX2, TNAP, ENPP1, COL10A1,* and *MKX*, and the absence of the expression of *COL2A1* (Figure 4B). TLC in chondrogenic pellets expressed higher levels of chondrocyte markers *COL10A1* and *MMP13* than TLC in two-dimensional (2D) cultures. We also observed the re-expression of the tenocyte marker *TNMD* in chondrogenic pellets, although it remained undetected in the control 2D condition.

Next, TLC 3D pellets were cultured for one week in chondrogenic medium, followed by three weeks in a mixed medium composed of the chondrogenic medium supplemented with osteogenic molecules (CH-OM). In this condition, we observed strong mineral deposits in the fibrocartilaginous peripheral tissue as well as in the poorly organized center of the pellets for only one in three donors (Figure 5A, donor 1). Calcium deposits appeared as small micrometric vesicles that eventually aggregated to form larger, amorphous deposits. No mineralization was observed for donors 2 and 3, despite the presence of osteogenic molecules (Figure 5A). When adding the TNAPi to the mixed medium (CH-OM + TNAPi), mineralization was totally prevented for donor 1 (Figure 5A). Cells located within the fibrocartilaginous alcian blue positive area were positive for TNAP in the chondrogenic and chondrogenic/osteogenic medium, but its expression was reduced in the presence of the TNAPi (Figure 5C).

Histological and 2D in vitro experiments showed us that a chondrocyte transition of TLC was associated with apatite deposition. In the last part, we studied the composition of calcifications extracted from patients to identify which components, possibly regulatory, were associated with crystals.

### 3.6. Composition of Calcium Deposits

Electron microscopy performed on the calcifications extracted from patients with calcific tendonitis showed a powder composed of ellipsoidal objects ranging from 3 to 300 µm (Figure 6A). FTIR spectroscopy found that the spectrum of apatite and additional peaks were observed between 1400–1700 cm^−1^ and between 2900–3500 cm^−1^ (Figure 6B). They were linked to the presence of proteins. 

Following extraction, 348 proteins were reliably identified by mass spectrometry-based proteomic analyses (Appendix A). Bioinformatic analysis indicated that the majority of proteins belong to the extracellular space, and many have been described as components of extracellular vesicles (Figure 6C). Considering the involvement in biological processes, we observed a significant enrichment of proteins involved in extracellular matrix organization, but also, more interestingly, of proteins involved in ossification, bone growth, and cartilage development (Figure 6C). This prompted us to compare our list of calcification-contained proteins to those reported to be present in connective tissues. For this, we compiled proteomic datasets previously reported to produce three lists of entries for (i) tendon [22,23,24,25,26], (ii) bone [27,28], and (iii) cartilage [29,30]. As the compiled and compared datasets were acquired from samples of different species, we relied on the gene names extracted from Uniprot [31] to perform the comparison between sample types. This analysis confirmed the presence of proteins already found in tendons, but also in bone and/or cartilage (some of them are reported in Figure 6D). Interestingly, we identified several proteins, such as ENPP1, NT5E (CD73), SERPINF1 (PEDF), or POSTN, which are known enzymes or regulating factors implicated in the mineralization process [32,33,34].

## 4. Discussion

To explore the mechanisms involved in the pathogenesis of calcific tendonitis, we collected three types of human samples: calcified tendon samples from cadaveric subjects, tenocyte-like cells from donors, and calcifications from patients with shoulder pain. The histological analysis of calcified tendons showed a fibrocartilaginous metaplasia containing chondrocyte-like cells around the deposits with adjacent blood vessels. This metaplasia has been previously described by Uhthoff [35]. While Sox9 was expressed, we did not detect any expression of collagen II in immunohistochemistry. This lack of collagen II has already been observed in calcific tendonitis [3]. While fibrocartilage is typically characterized by the presence of type II collagen, Benjamin et al. reported that some types of fibrocartilage contain only a small quantity of type II collagen and mainly type I collagen (knee joint menisci) [36]. Cells within these areas expressed RUNX2 and SOX9, which are transcription factors involved in chondrocyte differentiation. Some of them also expressed COL10, suggesting a hypertrophic phenotype, which is consistent with their expression of RUNX2 and TNAP [37]. During differentiation in hypertrophic chondrocytes, the expression of SOX9 and type II collagen is reduced before disappearing in parallel with an increase in production of type X collagen [38]. In the growth plate, the downregulation of SOX9 is essential for allowing cartilage vascularization and endochondral ossification [39]. Although we observed several common features to endochondral ossification (the expression of several hypertrophic chondrocyte markers and increased vascularization), several elements suggest a different mechanism: the maintenance of expression of SOX9, the absence of apoptosis of hypertrophic chondrocytes, the absence of bone structure and osteoblasts cells.

We next showed that cells extracted from rotator cuff tendons, called TLC, were able to induce mineral deposits in vitro. Our TLC had a fibroblastic morphology, and expressed tendon-specific and MSC markers, as previously reported [6,7,14]. The cells were used until passage 2, because the multiple passaging of TLC leads to phenotypic drift [40]. In previous studies, mineralization was relatively weak with TLC [41]. Interestingly, when using DMEM to expend TLC, we similarly did not observe any significant mineralization in osteogenic medium (data not shown). Therefore, the expansion of TLC in RPMI medium with low glucose appears to be a key step for obtaining cells with significant mineral deposit ability. Interestingly, the phenotype of the mineral-forming cells showed characteristics of hypertrophic chondrocytes, such as an increased expression of *TNAP, COMP*, *COL10,* and *MMP13*. *MMP13* plays a key role in chondrocyte hypertrophy and mineralization. Its transcription is regulated by *RUNX2*, and inhibiting *MMP13* can suppress hypertrophy and the expression of COL10. In addition, the inhibition of *MMP13* inhibits calcium incorporation in the extracellular matrix [42]. These experiments have two main limitations regarding the origin of the cells. First, TLC was extracted from the supraspinatus tendon or biceps tendon. However, it has been previously shown that TLC from either supraspinatus or biceps origins have comparable expression patterns [14]. Secondly, we used TLC extracted from OA donors. This could have influenced our results, because the mild inflammatory environment induced by OA could have had an impact on TLC phenotype and ability to induce mineralization, as shown for the anterior cruciate ligament cells in case of knee OA [43].

To explore whether or not cells with a cartilaginous phenotype were also able to mineralize, we cultured TLC in chondrogenic 3D pellets and exposed them to an osteogenic medium. These cultures led to inconstant mineral deposits. For donor 1 (Figure 5), striking similarities were noted with histological images obtained from calcified tendons, such as micrometric calcified vesicles and the presence of hypertrophic chondrocytes expressing TNAP. Donor 1 was the oldest of our donors (86 years), but did not present any other specific traits. A higher age could have influenced the results as shown by Mc Beath et al. Indeed, in their study, aged tenocytes had a higher propensity than young tenocytes to produce mineral deposits under hypoxic culture conditions [44].

We have shown the key role played by TNAP in calcification formation. TNAP is a key enzyme involved in the physiological mineralization process: extracellular inorganic pyrophosphates are provided by ENPP1 and then hydrolyzed by TNAP to promote mineralization. These two enzymes were expressed by hypertrophic chondrocyte-like cells in situ. In addition, we observed extracellular TNAP on the surface of the calcium deposits. We can speculate that TNAP is present in matrix vesicles derived from the hypertrophic chondrocyte-like cells, and that mineral deposits occur in these matrix vesicles. The presence of matrix vesicles has previously been observed in tendons by electron microscopy [3], and their potential implication in other pathological calcifications (vascular and cartilage calcifications) has been suggested [45]. In addition, TNAP expression was increased in TLC after 21 days in the osteogenic medium. Importantly, the inhibition of TNAP was able to decrease mineral deposits in vitro in 2D cultures. The TNAP inhibitor also reduced expression of the hypertrophic chondrocyte markers *COL10* and *MMP13*. This suggests that the TNAPi acted in part on mineralization by modifying the phenotype of the cells. We still need to determine whether TNAP inhibitors will be useful in the treatment of calcifying tendonitis, as suggested previously for vascular calcifications [46].

Finally, identification by mass spectrometry-based proteomics of the proteins associated with apatite crystals revealed an enrichment in cartilage-associated protein and the enzymes involved in cartilage metabolism that corroborates the previous results. It will be interesting in the future to study the role of these elements in the pathogenesis of the disease.

In conclusion, we provide here evidence that tenocytes have a propensity to differentiate or transdifferentiate into hypertrophic chondrocyte-like cells to form fibrocartilaginous tissue, and to produce TNAP-dependent calcium deposits. We believe these results may pave the way for further studies to identify the local regulating factors that may represent valuable targets for treating calcific tendonitis.

## Figures and Tables

**Figure 1 jcm-08-01544-f001:**
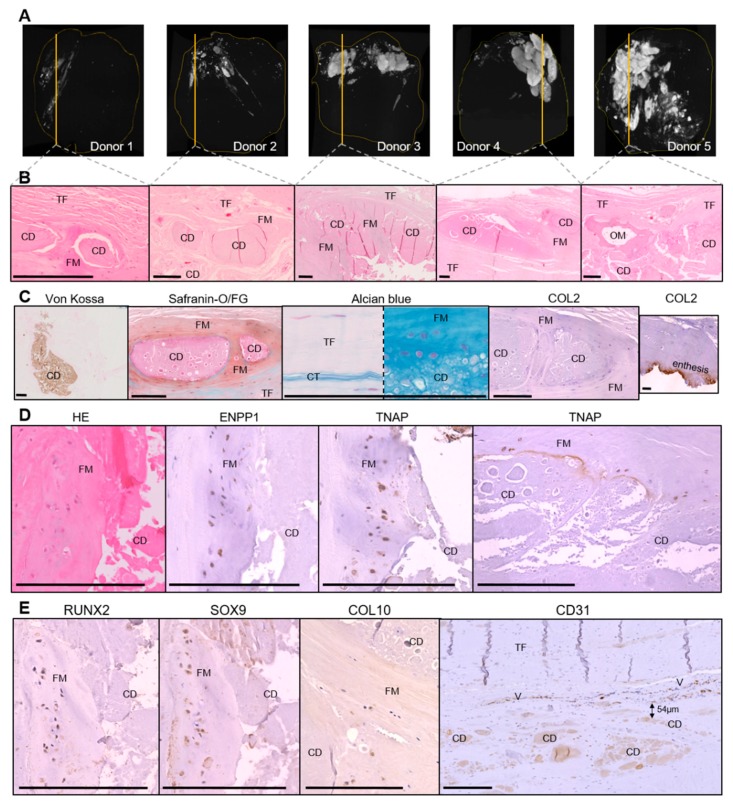
Histological characterization of rotator cuff calcifications (*N* = 5). (**A**) Calcific deposits assessed by micro-computed tomography. (**B**) HE (hematoxylin and eosin) staining of decalcified samples (1 image for each patient). (**C**) Von Kossa staining on not decalcified samples and characterization of the fibrocartilaginous area with alcian blue staining, SOFG (Safranin O/Fast Green) staining, and immunohistochemical staining for COL2 (brown) with Gill hematoxylin counterstain. (**D**,**E**) Representative immunohistochemical staining for ENPP1 (ectonucleotide pyrophosphatase/phosphodiesterase 1), TNAP (tissue non-specific alkaline phosphatase), Runx2, Sox9, Col10 and CD31 (brown). TF: tendon fibers; CD: calcium deposits, FM: fibrocartilaginous metaplasia; OM: osseous metaplasia; CT: connective tissue; V: vessels. Scale bar = 200 µM.

**Figure 2 jcm-08-01544-f002:**
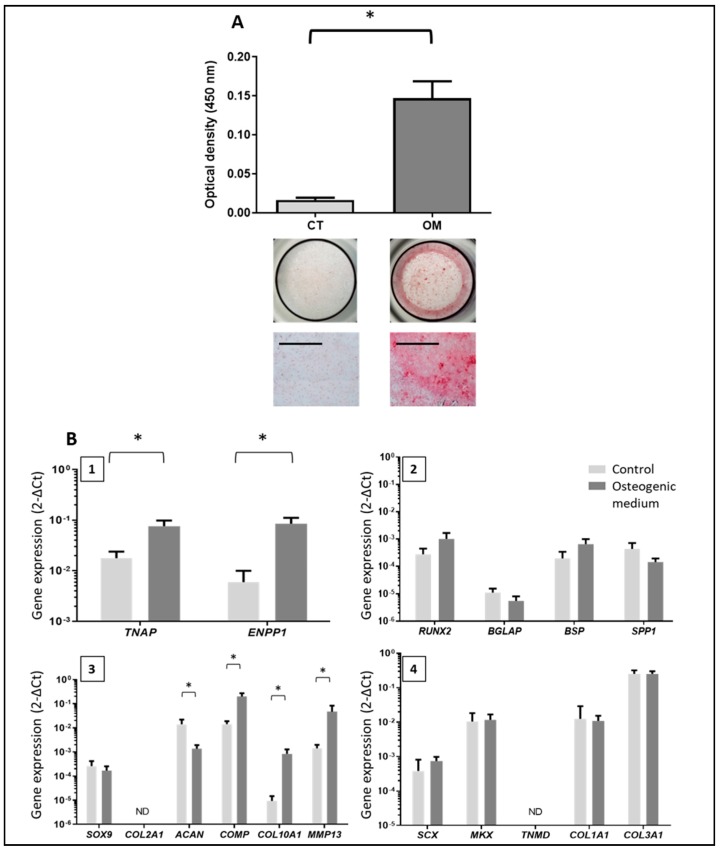
Assessment of mineral deposition induced by tenocyte-like cells (TLC) after 21 days in the osteogenic medium and characterization of mineralizing cells. (**A**) Assessment of mineral deposition by alizarin red S staining after 21 days (*N* = 5). Quantification of the bound stain after solubilization with formic acid for the two conditions (reading at 450 nm) and representative images of the wells. CT: control medium; TLC: tenocyte-like cells; OM: osteogenic medium. Scale bar = 1 mm. (**B**) Gene expression by TLC cultured 21 days in the osteogenic medium or in the control medium (N = 3 to 5). 1: Study of TNAP (tissue non-specific alkaline phosphatase) and ENPP1 (ectonucleotide pyrophosphatase/phosphodiesterase 1) levels of expression. 2: Study of osteoblast markers: RUNX2, BGLAP (bone gamma-carboxyglutamate protein), BSP (Bone SialoProtein) and SPP1 (secreted phosphoProtein 1). 3: Study of chondrocyte markers: SOX9, COL2A1, ACAN (aggrecan), COMP (cartilage oligomeric matrix protein), COL10A1, and MMP13 (matrix metallopeptidase 13). 4: Study of tenocyte markers: SCX (scleraxis), MKX (Mohawk homeobox), TNMD (tenomodulin), COL1A1 and COL3A1. * = *p* < 0.05.

**Figure 3 jcm-08-01544-f003:**
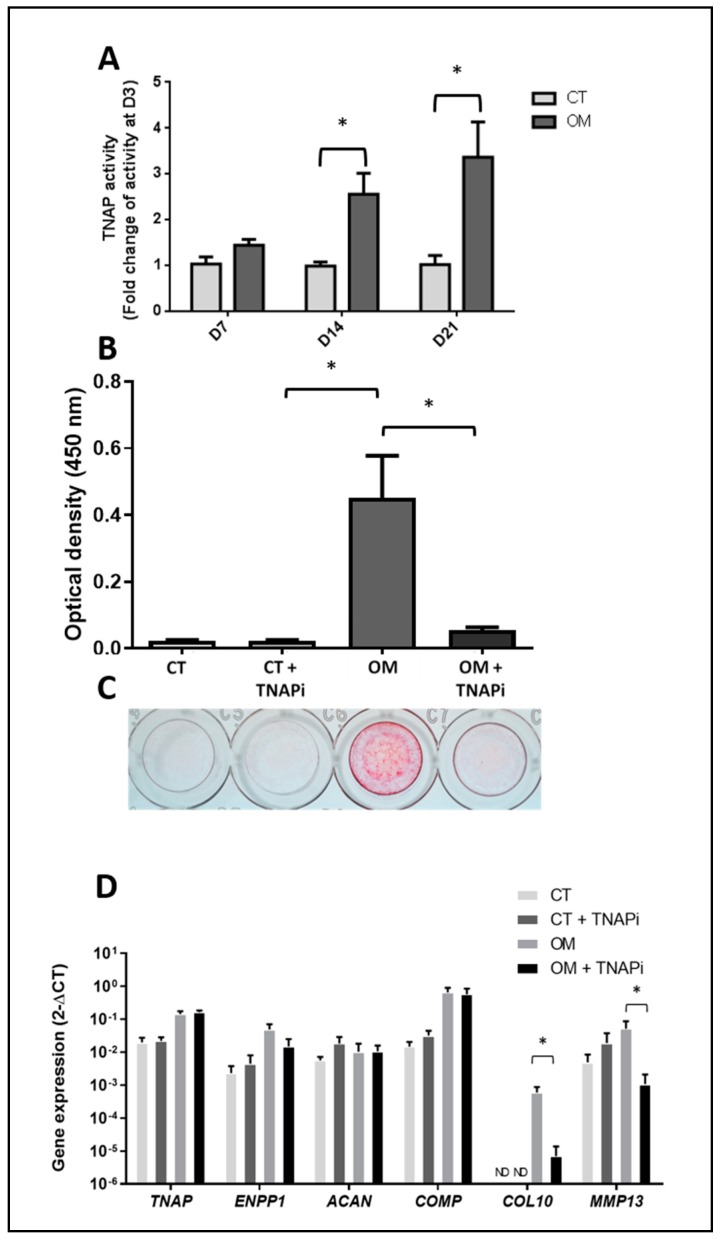
**Study of TNAP involvement in the mineralization process.** (**A**) TNAP enzymatic activity over time for the two conditions (*N* = 3). (**B**) Effects of a TNAP inhibitor on the mineral deposition (*N* = 3) assessed by the colorimetric method after the solubilization of alizarin red. (**C**) Representative images of alizarin red staining of wells of one patient. CT: control medium; OM: osteogenic medium; TNAPi: TNAP inhibitor. (**D**) Gene expression by TLC after treatment with a TNAP inhibitor (*N* = 3). TNAP: tissue non-specific alkaline phosphatase; ENPP1: ectonucleotide pyrophosphatase/phosphodiesterase; ACAN: aggrecan; COMP: cartilage oligomeric matrix protein; COL10: type X collagen; MMP13: matrix metallopeptidase 13; CT: control; OM: osteogenic medium; TNAPi: TNAP inhibitor; ND: not detected. * = *p* < 0.05.

**Figure 4 jcm-08-01544-f004:**
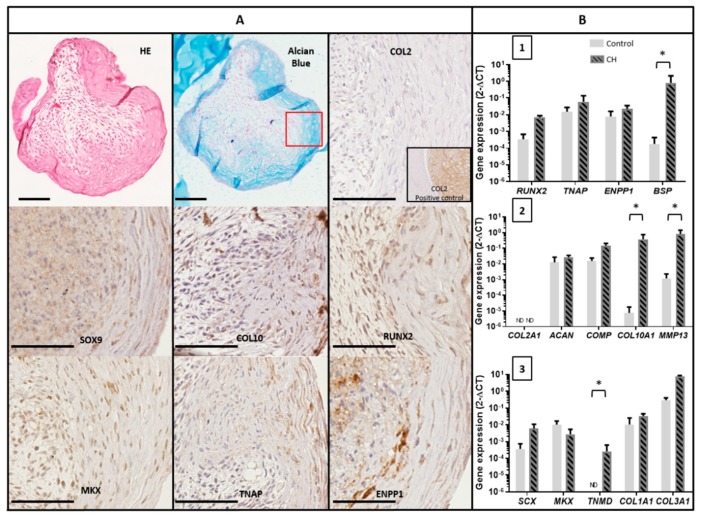
**Characterization of TLC cultured in chondrogenic pellets**. (**A**) Representative images of histological and immunohistochemical staining of TLC pellets (CH medium) for COL2, SOX9, COL10, RUNX2, MKX, TNAP, and ENPP1. COL2 positive control was obtained with a chondrosarcoma sample. HE: Hematoxylin and eosin. Scale bar 100 µm. (**B**) Gene expression of TLC cultured four weeks in chondrogenic pellets or in cell monolayer in control medium: levels of expression of osteoblast markers (1), chondrocyte markers (2) and tenocyte markers (3) (*N* = 3) assessed by qPCR.* = *p* < 0.05.

**Figure 5 jcm-08-01544-f005:**
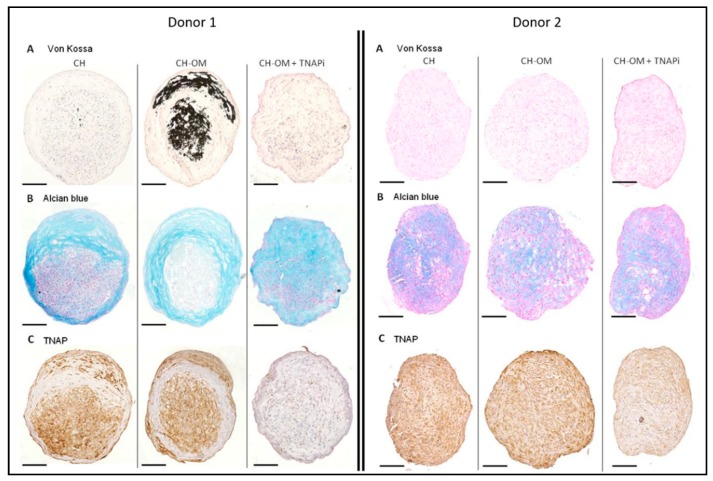
Mineralization ability of TLC in 3D pellets and effects of a TNAP inhibitor (*N* = 3). (**A**) Representative images of Von Kossa staining of TLC pellets for two patients: mineralization was observed with donor 1, contrary to donors 2 and 3. (**B**) Alcian blue staining. (**C**) TNAP staining. CH = chondrogenic medium for 4 weeks; CH-OM = chondrogenic medium for 1 week followed by 3 weeks in mixed chondrogenic/osteogenic medium; TNAPi = tissue non-specific alkaline phosphatase inhibitor, added for the last three weeks. Scale bar = 100 µm.

**Figure 6 jcm-08-01544-f006:**
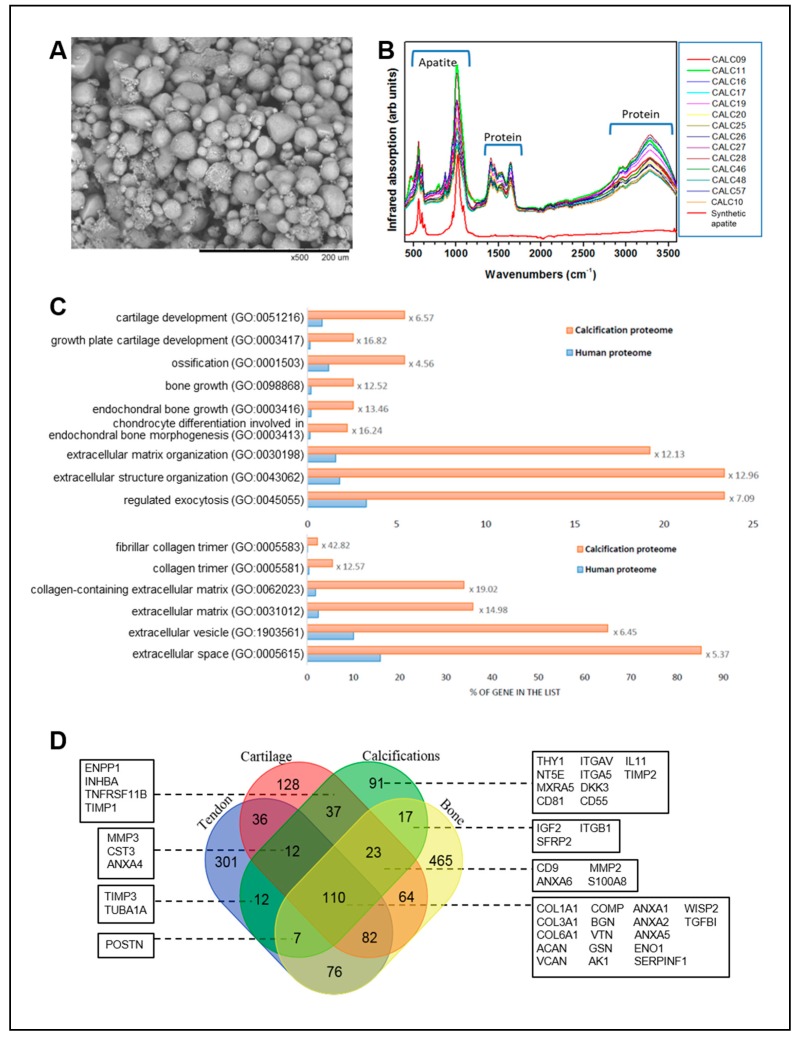
**Analysis of the composition of calcifications extracted from patients**. (**A**) Electron Microscopy (SEM) of the calcic powder extracted from a representative patient. Scale bar 200 µm. (**B**) Fourier transform infrared (FTIR) spectroscopy of the patients’ calcifications (*N* = 14). SHA = synthetic hydroxyapatite. (**C**) Bioinformatic analyses of the protein list identified in calcifications by mass spectrometry-based proteomic analyses (*N* = 5). Percentage of genes classified in selected statistically significant GO terms in the total human proteome (blue bars) and in calcifications (orange bars) for biological process (upper panel) and cellular components (lower panel). Enrichment is indicated for each term. (**D**) Comparison of the protein lists compiled for calcifications, bone, tendon, and cartilage using Venn diagram (http://bioinformatics.psb.ugent.be/webtools/Venn/). Selected proteins of interest are presented.

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
