# Peer review of "Rotator Cuff Tenocytes Differentiate into Hypertrophic Chondrocyte-Like Cells to Produce Calcium Deposits in an Alkaline Phosphatase-Dependent Manner"

_jcm, 2019, doi:10.3390/jcm8101544_

Round 1

Reviewer 1 Report

General comments

This manuscript presents a very interesting data regarding the possible mechanism of calcific tendonitis. As the authors mentioned, calcific tendonitis is a frequent cause of shoulder pain, but its detailed mechanism is still unknow and there is no way to prevent. Generally, heterotopic calcium deposit is caused by high level of serum phosphorus, so the increase of blood vessel and serum phosphorus should be investigated. Actually, this paper could not reach the goal to resolve the all pathogenesis of calcific tendonitis, but this includes lots of new insights. However, there are some issues remain to be properly explained and revised.

These are itemized below:

1)            As mentioned above, the increase of blood vessel and serum phosphorus should be investigated or discussed.

2)            The authors have investigated 5 cadaveric subjects and showed some characteristic findings (Figure1). However, this would be like case series, not a scientific paper, as it is. This reviewer can’t help but say that there is no scientific basis with the lack of statistical analysis.

3)            In figure 5, mineral deposit was observed in only 1 donor from 3 donors. This could be an accidental result, therefore this reviewer would like to see a scientific evidence.

4)            Line 64. Please indicate the approval number from the Ethics committee.

Author Response

Response to Reviewer #1 comments:

This manuscript presents a very interesting data regarding the possible mechanism of calcific tendonitis. As the authors mentioned, calcific tendonitis is a frequent cause of shoulder pain, but its detailed mechanism is still unknow and there is no way to prevent. Generally, heterotopic calcium deposit is caused by high level of serum phosphorus, so the increase of blood vessel and serum phosphorus should be investigated. Actually, this paper could not reach the goal to resolve the all pathogenesis of calcific tendonitis, but this includes lots of new insights. However, there are some issues remain to be properly explained and revised.

These are itemized below:

Point 1:   As mentioned above, the increase of blood vessel and serum phosphorus should be investigated or discussed.

Response to point 1: We thank the reviewer for this comment. Calcific tendonitis and heterotopic calcium deposit have different pathophysiology. For instance, among patients with shoulder calcific tendonitis, serum phosphorus is always normal and deposits always occur inside the tendon [Speed et al, 1999]. In contrast, heterotopic calcifications can be associated with high level of serum phosphorus, especially inpatients with renal impairment but these calcifications are quite different, usually more extensive and involving other tissues than the tendons (sub-cutaneous, muscles).

We added line 46: “In calcific tendonitis, deposits are not linked to dysregulation of calcium and phosphorus metabolism [4] as opposed to those observed in patients with chronic kidney disease. In the latter, heterotopic calcifications are more extensive and involve other tissues than the tendons (sub-cutaneous, muscles)”.

The following reference has been added: Speed, C.A.; Hazleman, B.L. Calcific tendinitis of the shoulder. N Engl J Med 1999, 340(20), 1582-4.

We now discuss more the increase of blood vessels line 342 and 353 in the discussion section: “Although we observed several common features to endochondral ossification (expression of several hypertrophic chondrocyte markers and increased vascularization), several elements suggest a different mechanism: maintenance of expression of SOX9, absence of apoptosis of hypertrophic chondrocytes, absence of bone structure and osteoblasts cells”.

Point 2: The authors have investigated 5 cadaveric subjects and showed some characteristic findings (Figure1). However, this would be like case series, not a scientific paper, as it is. This reviewer can’t help but say that there is no scientific basis with the lack of statistical analysis.

Response to point 2: Indeed, the first part of the result is mainly descriptive without statistical analysis but this histological study was an initial approach to describe the calcifications and their environment. Even if we did not perform any statistical tests, we systematically compared the non-calcified part of each tendon to the calcified one. No fibrocartilaginous metaplasia was observed in the non-calcified area and cells did not express fibrocartilaginous markers. In addition, “normal tenocytes” did not express markers such as Runx2, Sox9, TNAP, ENPP1 or type X collagen although chondocyte-like cells were positive within the same sample.

We have added line 178: “This fibrocartilaginous metaplasia was not observed within the non-calcified part of the samples or in the normal tendon” and line 199: “In contrast, tenocytes observed in the non-calcified part of the samples and in the normal tendon did not express these markers”.

Point 3: In figure 5, mineral deposit was observed in only 1 donor from 3 donors. This could be an accidental result, therefore this reviewer would like to see a scientific evidence.

Response to point 3: This is an important point. In the light of the results obtained in 2D cultures, we wanted to explore the effect of a heightened chondrogenic transition to obtain a more important mineralization. However, as discussed lines 373-380, cultures of TLC previously differentiated in chondrogenic medium led to inconstant mineral deposits. The same protocol was repeated three times and it needed lots of cells with a long-term culture (firstly to obtain a sufficient number of cells (more than 1.5 millions) and secondly to differentiate the cells in two steps). So when we observed that mineralization was not constant, we did not repeat it again. The donor 1 was the oldest of donors (86 years). This could have influenced the results because, as reported by McBeath et al., aged tenocytes had a higher propensity to produce mineral deposits particularly when cultured in hypoxia.

Although these results did not present scientific evidence, we thought it could be interesting to include them in our study because they suggest that a prior chondrogenic differentiation of TLC is not necessary to obtain hypertrophic chondrocyte-like cells or mineralization.

We added in the discussion line 384: "Donor 1 was the oldest of our donors (86 years) but did not present any other specific traits. A higher age could have influenced the results as shown by Mc Beath et al. Indeed, in their study, aged tenocytes had a higher propensity than young tenocytes to produce mineral deposits under hypoxic culture conditions [44]".

The following reference has been added: McBeath, R.; Edwards, R.W.; O'Hara, B.J.; Maltenfort, M.G.; Parks, S.M.; Steplewski, A.; Osterman, A.L.; Shapiro, I.M. Tendinosis develops from age- and oxygen tension-dependent modulation of Rac1 activity. Aging Cell 2019, 18(3), e12934.

Point 4: Line 64. Please indicate the approval number from the Ethics committee.

Response to point 4: A local approval has been obtained in the anatomy institute to use cadaveric subjects for scientific studies. However, there is no number associated with this approval.

Reviewer 2 Report

Calcific tendonitis is a frequent cause of chronic shoulder pain. To explore the mechanisms involved in the pathogenesis of calcific tendonitis, the authors collected three types of human samples: calcified tendon samples from cadaveric subjects, tenocyte-like cells from donors and calcifications from patients with shoulder pain. (A) The infra-red spectroscopy and mass spectrometry-based proteomic characterizations were performed and mineralization ability of human rotator cuff cells was assessed. (B) The histological analysis of calcified tendons showed a fibrocartilaginous metaplasia containing chondrocyte-like cells around the deposits.
Important findings: (A) Calcifications were amorphous areas surrounded by a fibrocartilaginous metaplasia containing hypertrophic chondrocyte-like cells that expressed Tissue Non-specific Alkaline Phosphatase (TNAP) and Ectonucleotide Pyrophosphatase/Phosphodiesterase 1 (ENPP1). Calcific deposits were composed of apatite crystals associated with proteins involved in bone and cartilage development and endochondral bone growth. Although metaplasia has been previously described, and lack of collagen II has already been observed in calcific tendonitis, the authors found that cells expressed RUNX2 and SOX9, are transcription factors involved in chondrocyte differentiation. They found in the growth plate, the downregulation of SOX9 is essential for allowing endochondral ossification. They proposed that hypertrophic chondrocytes did not reach their terminal differentiation. (B) They also noted that in vitro, the use of a TNAP inhibitor significantly prevented mineral deposits.
Strength: This study provides evidences that tenocytes have a propensity to differentiate into hypertrophic chondrocyte-like cells to produce TNAP- dependent calcium deposits. These results may pave the way to identifying regulating factors that might represent valuable targets in calcific tendonitis.
Conclusion: this is a well-conducted study, and a well-written paper, providing evidence of tenocyles’ propensity to differentiate into hypertrophic chondrocyte-like cells and the resulting calcium deposits. I suggest to accept after minor revision for some typos.

Author Response

Response to reviewer#2: We would like to thank the reviewer for the positive comments. The whole manuscript has been checked by three different persons (CDL, BLG and FB) and corrected for typographic errors.

Reviewer 3 Report

Title: Rotator cuff tenocytes differentiate into hypertrophic chondrocyte-like cells to produce calcium deposits in an alkaline phosphatase-dependent manner

By Darrieutort-Laffite et al.,

Calcium deposition can be observed in diverse ligamentous and cartilaginous tissues and remains a matter of debate with mostly unknown pathogenesis. Hence, Darrieutort-Laffite et al., present an interesting manuscript whith a novel perspective. Strengness of the manuscript is the combination of clinical patient data, in silico proteomics and in vitro analysis. One limitation is the inclusion of only one healthy donor as control and the integration of several biceps samples instead of supraspinatus tissues for the in vitro analyses. The exclusion of isolating synovial fibroblasts from this intrasynovial tendon should be considered. The latter were exposed to OA – this fact should be included into the discussion of data, since tenocytes and ligamentocytes are affected by OA.

Abstract: line 19: „In vitro, the mineralization ability of human rotator cuff cells was assessed.“ Please add here which method was used and whether healthy or affected tissue was investigated.

Page 2, Line 46: „maintenance of connective tissue“ please add: „dense regular connective tissue“

Line 50: „Mohawk“ should be added, since it was also analyzed (see later)

As additional reference for calcinosis and OA Ruschke et al., 2016 (PMID27208419)

referring to a similar feature in the ACL could be integrated.

Line 67: please add the gender of the donors

2.2 cell culture

Line 83: the „Biceps“ does not belong to the rotator cuff. In contrast to the supraspinatus tendon (supraspinatus: please write it wihout „-„) it is an intrasynovial tendon. Please add also the gender of the donors. The samples used for cell isolation derived from OA – were these samples also checked for the presence of calcinosis before collagenase digestion? The influence of the inflammatory milieu associated with OA on tenocytes should be considered/discussed.

Line 98: a blank is lacking.

Page 3, line 96: Cell pellets were obtained following the manufacturer’s instructions: please describe the method to achieve pellet formation

Line 153: „acellular material“ do the cells die (they could die without caspase-3 activation)? This fact is interesting – in contrast to bone formation by endochondral osteogenesis no cells survive here?

Line 165: „on no decalcified samples“ better to write „not decalcified“

Line 175: „eventually“ what does it mean? Not all samples positive?

Line 197: something is missing?

Line 243: „The center of the pellets was much less structured, with poor extracellular matrix.“ Diameter of the pellets? Possibly a consequence of malnutrition?

3.5

Sox9, a transcription factor which regulates collagen type II is expressed, but not type II collagen? Please comment! How about the protein expression (fig. 4) of the cartilage marker aggrecan which is also regulated by sox9? (see also line 323, discussion section)

Line 259: please explain the rationale for the experimental setting and time course of analyses?

Discussion

Iine 338-341: „MMP13 vs. MMP-13“ please select a consistent writing style

line 342: which donor specific traits might influence this?

Authors contribution and funding is lacking

Line 371: comment the local milieu.

Author Response

Point 1:Strengness of the manuscript is the combination of clinical patient data, in silico proteomics and in vitro analysis. One limitation is the inclusion of only one healthy donor as control and the integration of several biceps samples instead of supraspinatus tissues for the in vitro analyses. The exclusion of isolating synovial fibroblasts from this intrasynovial tendon should be considered. The latter were exposed to OA – this fact should be included into the discussion of data, since tenocytes and ligamentocytes are affected by OA.

Response to point 1: We want to thank the reviewer for the positive comment about the strengths of our research.

We recognize that the inclusion of only one healthy donor is a limitation of the analysis. However, the not calcified parts of the calcified tendons were also considered as controls. In this part of the samples, we did not observe the fibrocartilaginous metaplasia and the chondrocytes like cellsas in the normal tendon. We also observed that “normal tenocytes” did not express markers such as Runx2, Sox9, TNAP, ENPP1 or type X collagen although chondocyte-like cells were positive within the same sample.

We added line 178: “This fibrocartilaginous metaplasia was not observed within the not calcified part of the samples or in the normal tendon” and line 199: “In contrast, tenocytes observed in the not calcified part of the samples and in the normal tendon did not express these markers”.  

As pointed out by the reviewer, the use of cells derived from two different tendons could represent a limitation. Biceps is an intrasynovial tendon and our cell cultures could have been “contaminated” by synovial fibroblasts. For cell extraction from biceps or supraspinatus, we used a procedure previously published [14]. In this study, tenocyte-like cells from either supraspinatus or biceps origin have comparable expression patterns for all analyzed markers. In our experiments, we did not observe any difference in the mineralization obtained in vitro regardless of the origin of TLC. However, we cannot exclude that a little proportion of synovial cells could have been present in our cultures.

The second limitation about in vitro results is that tenocytes used in our experiments were extracted from donors suffering from OA. This may have influenced cell behavior or phenotype. To avoid impact of an inflammatory environment, we excluded donors with a diagnosis of rheumatoid or others inflammatory arthritis. However, it is true that we cannot exclude an impact of a mild inflammatory reaction in the context of OA. But in fact, for ethical reasons, obtaining tenocytes from normal shoulder is not currently feasible.

The response to these two comments was added line 372: "These experiments have two main limitations regarding the origin of the cells. First, TLC were extracted from supraspinatus tendon or biceps tendon. However, it has been previously shown that TLC from either supraspinatus or biceps origin have comparable expression patterns [14]. Secondly, we used TLC extracted from OA donors. This could have influenced our results because the mild inflammatory environment induced by OA could have had an impact on TLC phenotype and ability to induce mineralization as shown for anterior cruciate ligament cells in case of knee OA [43]".

The following reference has been added : Ruschke, K.; Meier, C.; Ullah, M.; Krebs, A.C.; Silberreis, K.; Kohl, B.; Knaus, P.; Jagielski, M.; Arens, S.; Schulze-Tanzil, G. Bone morphogenetic protein 2/SMAD signalling in human ligamentocytes of degenerated and aged anterior cruciate ligaments. Osteoarthritis Cartilage 2016, 24(10), 1816-1825.

Point 2: Abstract: line 19: „In vitro, the mineralization ability of human rotator cuff cells was assessed.“Please add here which method was used and whether healthy or affected tissue was investigated.

Response to point 2:

Line 20, we provided details about donors and methods used to assess mineralization : “In vitro, the mineralization ability of human rotator cuff cells from osteoarthritis donors was assessed by alizarin red or Von Kossa staining”.

Point 3:Page 2, Line 46: „maintenance of connective tissue“ please add: „dense regular connective tissue“

Response to point 3:

Line 51, we completed the sentence as follows :“maintenance of dense regular connective tissue”.

Point 4:Line 50: „Mohawk“ should be added, since it was also analyzed (see later)

Response to point 4: l.53, we added “Mohawk homeobox”

Point 5:As additional reference for calcinosis and OA Ruschke et al., 2016 (PMID27208419)

referring to a similar feature in the ACL could be integrated.

Response to point 5: this point has been discussed in the response to point 1.

Point 6:Line 67: please add the gender of the donors

Response to point 6: l.70, we added “3 males”

2.2 cell culture

Point 7: Line 83: the „Biceps“ does not belong to the rotator cuff. In contrast to the supraspinatus tendon (supraspinatus: please write it wihout „-„) it is an intrasynovial tendon. Please add also the gender of the donors. The samples used for cell isolation derived from OA – were these samples also checked for the presence of calcinosis before collagenase digestion? The influence of the inflammatory milieu associated with OA on tenocytes should be considered/discussed.

Response to point 7:

This is challenging question. The long head of the biceps does not belong to the group of rotator muscles of the shoulder. However, its tendon passes over the head of the humerus and is in close association with the rotator cuff. It is always included in the examination of the shoulder because it functions intimately with the rotator cuff as a humeral head depressor.In elevated positions, the long head of the biceps stabilized the joint anteriorly when the arm was internally rotated and stabilized the joint posteriorly when the arm was externally rotated [Terry, J Athl Train. 2000]. In a study previously cited [Pauly], tenocyte-like cells extracted from supraspinatus or biceps tendon were similar. However, this limitation was taken into account and discussed line 372 (cf response to point 1).

Prior to sample collection, tendons were checked for the absence of calcification by looking at the X-ray of the shoulder. We added line 93: “Prior to sample collection, the absence of tendon calcification was checked on the X-ray of the shoulder”.

Gender of the donors has been added line 91: “10 females”

OA environment has been discussed in the response to point 1.We also have added details about type of OA line 92: "6 for primitive OA, 4 for rotator cuff tear arthropathy and 1 for OA secondary to osteonecrosis of the humeral head".

Point 8:Line 98: a blank is lacking.

Response to point 8:a blank was added line 114

Point 9:Page 3, line 96: Cell pellets were obtained following the manufacturer’s instructions: please describe the method to achieve pellet formation

Response to point 9:

More details about methods to obtain pellets have been included line 108: "To obtain pellets, cells were detached from the culture flask with Trypsin-EDTA (Lonza) and after centrifugation, they were counted and resuspended to generate a cell solution of 1.5 x 107 cells/ml in StemPro Osteocyte/Chondrocyte Differentiation basal medium. Five-µl droplets of cell solution were seeded in the center of wells (24-well plate) and were then incubated during 2 hours, at 37°C, 5% CO2. After 2 hours, complete StemProChondrogenesis Differentiation Medium (A10064-01, Gibco, USA) was added and changed every 2-3 days during a week".

Point 10: Line 153: „acellular material“ do the cells die (they could die without caspase-3 activation)? This fact is interesting – in contrast to bone formation by endochondral osteogenesis no cells survive here?

Response to point 10: It is indeed an interesting point. No cells were observed within the calcified areas. We can speculate that cells responsible for the deposits stay alive in the peripheral zone and that deposits enlarge centrifugally. Indeed, this fact is in contrast to bone formation by endochondral transition during which osteoblast cells replace apoptotic hypertrophic chondrocytes to produce mature bone. We have added this point in the discussion.

Line 355, we have replaced "However, in the case of calcific tendonitis, we usually did not observed ossification suggesting that these hypertrophic chondrocytes did not reach their terminal differentiation" by: “Although we observed several common features to endochondral ossification (expression of several hypertrophic chondrocyte markers and increased vascularization), several elements suggest a different mechanism: maintenance of expression of SOX9, absence of apoptosis of hypertrophic chondrocytes, absence of bone structure and osteoblasts cells”.

Point 11: Line 165: „on no decalcified samples “ better to write „not decalcified“

Response to point 11: the term has been corrected line 185.

Point 12:Line 175: „eventually” what does it mean? Not all samples positive?

Response to point 12:in the different samples, we observed that some chondrocyte-like cells expressed type X collagen and others did not.

In the text, we have replaced “eventually” with “some of them expressed type X collagen” (line 195).

Point 13: Line 197: something is missing?

Response to point 13: the end of the sentence appeared just above Figure 2 on the next page (page layout problem). New line 219.

Point 14:Line 243: „The center of the pellets was much less structured, with poor extracellular matrix.“ Diameter of the pellets? Possibly a consequence of malnutrition?

Response to point 14: The mean diameter of chondrogenic pellets obtained from tenocytes was 439 mm (+/-54). In the following reference [Fernandes et al, Tissue Eng Part C Methods. 2018] using the same kit, pellets obtained from mesenchymal stem cells (MSC) measured about 400 µm diameter not suggesting malnutrition in our cultures. The disorganization of the center of the pellets could be explained by an impaired diffusion of the soluble factors present in the medium with limited effects, as we observed previously with chemotherapeutic agents [Monderer, Lab invest 2013].

See lines 264: “The mean diameter of the pellets was 439µm (+/- 54µm)” and lines 265: “This could be explained by an impaired diffusion of the soluble factors present in the medium with limited effects, as we observed previously with chemotherapeutic agents [21]”.

The following reference has been added: Monderer, D.; Luseau, A.; Bellec, A.; David, E.; Ponsolle, S.; Saiagh, S.; Bercegeay, S.; Piloquet, P.; Denis, M.G.; Lodé, L.; Rédini, F.; Biger, M.; Heymann, D.; Heymann, M.F.; Le Bot, R.; Gouin, F.; Blanchard, F. New chondrosarcoma cell lines and mouse models to study the link between chondrogenesis and chemoresistance. Lab Invest 2013, 93(10), 1100-14.

3.5

Point 15:Sox9, a transcription factor which regulates collagen type II is expressed, but not type II collagen? Please comment! How about the protein expression (fig. 4) of the cartilage marker aggrecan which is also regulated by sox9? (see also line 323, discussion section)

Response to point 15:

Thank you for this relevant comment. While Sox9 was expressed, we did not detect any expression of collagen II in 2D cultures as in immunohistochemistry. As notified, Sox9 is a potent activator of type II collagen expression in articular chondrocytes. However, other transcription factors are required for the expression of collagen II and aggrecan as Sox-5 and Sox6. In addition, considering the hypertrophic phenotype of our cells, a reduced expression of collagen II is expected in parallel with an increase in production of collagen X. In the case of calcific tendonitis, the lack of collagen II within the fibrocartilaginous tissue has yet been reported by previous studies (line 337).

To discuss this point, we added line 343: “While Sox9 was expressed, we did not detect any expression of collagen II in immunohistochemistry” and line 350 : "During differentiation in hypertrophic chondrocytes, expression of SOX9 and type II collagen is reduced before disappearing in parallel with an increase in production of type X collagen [38]".

The following reference has been added: Goldring, M.B.; Tsuchimochi, K.; Ijiri, K. The control of chondrogenesis. J Cell Biochem 2006, 97(1), 33-44.

And line 351: Although we observed several common features to endochondral ossification (expression of several hypertrophic chondrocyte markers and increased vascularization), several elements suggest a different mechanism: maintenance of expression of SOX9, absence of apoptosis of hypertrophic chondrocytes, absence of bone structure and osteoblasts cells.

Immunohistochemical staining of aggrecan (ACAN) has not been performed on pellets. Unfortunately, we no longer have samples to perform this staining. However, ACAN mRNA expression was down regulated in 2D osteogenic culture (fig2), and this decreased expression was expected for hypertrophic chondrocytes.

Point 16:Line 259: please explain the rationale for the experimental setting and time course of analyses?

Response to point 16: 3D experiments have been performed after the results of 2D cultures given the observed phenotype of the TLC in mineralizing condition. First, we have performed the experiments in basic chondrogenic condition (4 weeks) to determine if TLC would be able to form pellets and express more chondrogenic markers. After this step, we sought to determine whether adding the osteogenic medium would induce mineral deposition. We have chosen to combine the chondrogenic medium and the osteogenic molecules because without the chondrogenesis medium, cells re-adhered in a monolayer (data not shown). One week in chondrogenic medium was necessary to obtain well-formed pellets in culture, leaving the remaining 3 weeks of culture to add osteogenic molecules. 3 weeks (21 days) is also the necessary time to induce mineralization in 2D culture (fig2 and 3).

Details about rationale for the experimental settings and time course have been added line 117: "Chondrogenic medium was combined with the osteogenic molecules because without the chondrogenic medium, cells re-adhered in a monolayer (data not shown). One week in chondrogenic medium was necessary to obtain well-formed pellets in culture to which we added 3 weeks of culture with osteogenic molecules (same duration as 2D-culture)".

Discussion

Point 17:Iine 338-341: „MMP13 vs. MMP-13“ please select a consistent writing style

Response to point 17: The writing style MMP13 has been applied to the entire document

Point 18:line 342: which donor specific traits might influence this?

Response to point 18: Donor 1 was the oldest among our donors (86 years). We did not identified other specific traits (co-morbidities, treatments, type of OA). Interestingly, it has been shown in a previous study that aged tenocytes had a higher propensity than young tenocytes to produce mineral deposits under hypoxic culture conditions [Mc Beath et al, Aging cell 2019] so we can speculate that this parameter could have influence the mineralization. Obviously, it would have been interesting to repeat this experiment more than three times to identify particular features in "mineralizing" donors. However, it needed lots of cells with a long-term culture (firstly to obtain a sufficient number of cells (more than 1.5 millions) and secondly to differentiate the cells in two steps).

Line 384, we added: "Donor 1 was the oldest of our donors (86 years) but did not present any other specific traits. A higher age could have influenced the results as shown by Mc Beath et al. Indeed, in their study, aged tenocytes had a higher propensity than young tenocytes to produce mineral deposits under hypoxic culture conditions [44]".

The following reference has been added: McBeath, R.; Edwards, R.W.; O'Hara, B.J.; Maltenfort, M.G.; Parks, S.M.; Steplewski, A.; Osterman, A.L.; Shapiro, I.M. Tendinosis develops from age- and oxygen tension-dependent modulation of Rac1 activity. Aging Cell 2019, 18(3), e12934.

Point 19:Authors contribution and funding is lacking

Response to point 19: Contributions have been added : conceptualization and methodology, C.D.L, Y.C., V.T., B.L.G. and F.B.; formal analysis, C.D.L., P.A., A.A., Y.C., G.L., B.L.G. and F.B. ; investigation, C.D.L., P.A., T.G., A.A., Y.C. and V.T.;data curation, C.D.L., P.A., T.G., A.A., Y.C. and G.L.; writing—original draft preparation, C.D.L., Y.C., F.B. and B.L.G.; writing—review and editing, V.T. and P.L.; supervision, P.L.; funding acquisition, C.D.L., Y.C., F.B. and B.L.G”.

Sources of funding have been added: Proteomic experiments were partly supported by ProFI (ANR-10-INBS-08-01 grant) and the Labex GRAL (ANR-10-LABX-49-01 grant). This work was also supported by grants from the French Society for Rheumatology (SFR 2017-18/2 grant and SFR 2018-4057).

Point 120:Line 371: comment the local milieu.

Response to point 20: line 411, we have added "local regulating factors" instead of "regulating factors".

Round 2

Reviewer 1 Report

This revised version of the paper is much more valuable to read. The studies are done well and of intrinsic biological and potential therapeutic importance. This manuscript is also well-designed, clinically reasonable, and convincing in many points with appropriate references.

Reviewer 3 Report

Dear authors,

My comments have been sufficiently addressed. Hence, I recommend the manuscript for publication.